# A Review of the Mechanical Properties and Durability of Ecological Concretes in a Cold Climate in Comparison to Standard Ordinary Portland Cement-Based Concrete

**DOI:** 10.3390/ma13163467

**Published:** 2020-08-06

**Authors:** Ankit Kothari, Karin Habermehl-Cwirzen, Hans Hedlund, Andrzej Cwirzen

**Affiliations:** 1Building Materials, Department of Civil, Environmental and Natural Resources Engineering, Luleå University of Technology, 97187 Luleå, Sweden; karin.habermehl-cwirzen@ltu.se (K.H.-C.); hans.hedlund@ltu.se (H.H.); andrzej.cwirzen@ltu.se (A.C.); 2Skanska Teknik AB, Skanska Sverige AB, 40518 Göteborg, Sweden

**Keywords:** ordinary Portland cement (OPC), supplementary cementitious materials (SCM), alkali-activated concrete (AAC), chemical admixtures, sustainable concrete, mechanical properties, frost durability

## Abstract

Most of the currently used concretes are based on ordinary Portland cement (OPC) which results in a high carbon dioxide footprint and thus has a negative environmental impact. Replacing OPCs, partially or fully by ecological binders, i.e., supplementary cementitious materials (SCMs) or alternative binders, aims to decrease the carbon dioxide footprint. Both solutions introduced a number of technological problems, including their performance, when exposed to low, subfreezing temperatures during casting operations and the hardening stage. This review indicates that the present knowledge enables the production of OPC-based concretes at temperatures as low as −10 °C, without the need of any additional measures such as, e.g., heating. Conversely, composite cements containing SCMs or alkali-activated binders (AACs) showed mixed performances, ranging from inferior to superior in comparison with OPC. Most concretes based on composite cements require pre/post heat curing or only a short exposure to sub-zero temperatures. At the same time, certain alkali-activated systems performed very well even at −20 °C without the need for additional curing. Chemical admixtures developed for OPC do not always perform well in other binder systems. This review showed that there is only a limited knowledge on how chemical admixtures work in ecological concretes at low temperatures and how to accelerate the hydration rate of composite cements containing high amounts of SCMs or AACs, when these are cured at subfreezing temperatures.

## 1. Introduction

### 1.1. Ordinary Portland Cement (OPC)-Based Concrete

The performance of ordinary Portland cement concrete strongly depends on the properties of the cementitious binder and the exposure conditions. The hydration rate is one of the main factors affecting the strength development of concrete. At lower temperatures its magnitude is reduced, and the setting time is elongated and the strength development is decreased by 20–40% [1,2,3]. During freezing, when the temperature drops around or below −4 °C, the moisture migrates within the binder matrix and ice starts to form hindering hydration processes and phase conversions of ettringite to monosulfate, as seen in Figure 1a [4,5,6]. OPC-based paste exposed to a temperature of −5 °C showed a hydration rate of 16.7% after 1 day and continued to hydrate to approximately 63.2% after 90 days, which is significantly lower in comparison with specimens cured at 20 °C (91.9%). Despite the low temperature of −5 °C, the hydration continued since the ions Ca^2+^, K^+^, Na^+^, OH^−^, and SO_4_^2−^ present in the cement, which was dissolved in water, prevented ice formation to a certain extent [5]. However, in the solidified state, the formed ice expands by up to 9% in concrete causing significant stresses and strains on the pore walls of the concrete. This can lead to permanent damage if the tensile strength of the binder matrix is too low, as seen in Figure 1c [1,5,7,8,9,10,11].

Therefore, most concrete guidelines recommend the application of special preventive measures when the air temperature drops below 5 °C or when it is colder than 10 °C for at least 12 h during the day [7,10,12,13]. Typically, these steps include heating the mixing water or aggregates, covering with isolative blankets or adding heating systems to the formworks [14]. Another possibility is the optimization of the concrete mix by the rapid hardening of Portland cement and various types of chemical admixtures, which has the potential to develop higher hydration heats and lower the freezing point of water [1,15,16]. Apparently, this procedure simplifies concrete production in moderately cold climatic conditions without the need of any internal or external heating, isolative covers or tents [7,9,17,18,19]. However, to prevent early permanent damage, the concrete should reach a critical compressive strength of 3.5 MPa before or during exposure to frost conditions [13,20]. Therefore, concretes cast in winter conditions contain antifreeze and accelerating admixtures to lower the freezing point, increase the hydration rate, shorten the setting time and accelerate the strength development (Figure 1b). Initially, in the early 1950s, two types of admixtures were used, which included chloride and non-chloride [1,7,21,22]. However, full-scale applications revealed that chloride-based (calcium and sodium) chemical admixtures increased the corrosion of the steel reinforcement and promoted water uptake into the concrete [23]. Chlorides are also used to remove ice from roads and pedestrian paths which can cause the surface scaling of structures if exposed to freeze–thaw cycles [24]. Air-entrainment, producing uniformly distributed air-voids within the binder matrix, was observed to substantially limit this kind of deterioration (Figure 1d) [25,26]. Table 1 lists the most commonly used antifreeze admixtures for winter concreting.

### 1.2. SCM-Based Concrete

Recent decades showed an increased interest and use of new ecological binders containing large amounts of supplementary cementitious materials (SCMs) or alkali-activated binders. The commonly used SCMs include industrial byproducts such as blast furnace slag, fly ash, silica fume or limestone. However, these binders are known for developing a lower heat of hydration leading to a delayed setting, slower strength development and often long-term durability problems [49,50,51,52,53,54]. Some of the SCM (slag, fly ash) rich in alumina and silica (aluminosilicates) can be activated with strong alkalis. Composite concretes based on alkali-activated binders showed a lower freezing point for the mixing water. For example, an 18% NaOH solution decreased the freezing point to −20 °C [6]. Alkali-activated binders showed also a better resistance against chemical attacks, chloride penetration and freeze–thaw cycles compared to the OPC mix at sub-zero temperatures [53,54].

Not all chemical admixtures developed for use in OPCs are equally efficient for ecological binders. In some cases, they might even produce adverse effects. The main objective of this review was to analyze the applicability of currently available chemical admixtures for applications in composite concretes which are exposed to low and sub-zero curing temperatures. For a fast review, tables including the results and recommendations are provided at the end of this publication.

## 2. OPC-Based Concretes

Fresh and hardened concrete properties differ significantly depending on the used chemical admixtures. Calcium chloride has been known for decades as an antifreeze agent which not only accelerates strength development during cold weather, but can also improve the rheology. However, it also negatively impacts the corrosion of the steel reinforcement. Research has shown and proved that 2–4 wt% of CaCl_2_ appeared to be the most efficient amount for concreting at temperatures down to −7 °C and did not have any adverse effects on the fresh and hardened concrete properties [24,29,30,31].

Furthermore, the combinations of calcium chloride and sodium chloride (3% + 7%), (10% + 5%) and (15% + 7%) enabled concrete works at −10 °C by developing a compressive strength of 32 MPa, 17 MPa, and 16 MPa, respectively, after 28 days. The results indicate that the application of higher dosages than the optimum percentage of CaCl_2_ led to a decrease in the compressive strength due to combined effects of expansive crack formation and increased permeability [1,7,28]. Similarly, concentrated magnesium chloride or calcium chloride caused the formation of expansive oxychloride compounds and subsequent cracking of the binder matrix with a reduction in the compressive strength by 30–50% [32].

Besides the chloride-based admixtures, the emerging non-chloride-based calcium nitrate admixture also showed similar accelerating effects on the strength development with a temperature down to −10 °C when a concentration of 6% was used. No signs of corrosion on the steel were detected [20]. The research concluded that this admixture is as soluble in water as other calcium salts, but only acts as a set accelerator due to the early formation of Portlandite [55,56]. However, further decreasing the temperature down to −20 °C deteriorated the concrete due to microcracking, but additional water curing significantly enhanced the strength properties by reviving the microstructure. Thus, calcium nitrate is the most suitable for countries and regions experiencing cold weather and a freeze–thaw cycle (F–T) [7,20,27,33]. Karagöl et al. (2013) produced workable OPC containing 6 wt% calcium nitrate and 0.5 wt% of superplasticizer. Concretes cured at −5 °C, −10 °C, −15 °C and −20 °C for 28 days reached compressive strength values of 33.21 MPa, 10.76 MPa, 5.35 MPa and 4.13 MPa, respectively. However, an additional 28 days of water curing at room temperature enhanced the strength to 57.51 MPa, 52.93 MPa, 51.35 MPa and 47.02 MPa for the samples initially cured at −5 °C, −10 °C, −15 °C and −20 °C, respectively [20]. Research conducted by Arslan et al. (2011), justifies the use of 1% calcium nitrate or 1% polyhydroxy amine antifreeze admixtures in an OPC mix at different curing temperatures (0 °C, −5 °C, −10 °C, −15 °C and −20 °C) for a period of 2 days, followed by 26 days of a water curing at room temperature [46]. With the presence of calcium nitrate, the compressive strength developed between 23.24 MPa and 14.8 MPa and between 25.53 MPa and 15.98 MPa for polyhydroxy amine. Similar results can be found for concrete using a calcium nitrite admixture cured at −4 °C for 3 days, followed by 25 days of further curing at 10 °C. It was seen that the setting time was shortened by six hours and that the 3-day compressive strength increased by 170% and the 28-day compressive strength increased by 117% compared to normal concrete with no admixtures but with the same curing regime [22]. Another study assessed an admixture containing a combination of polyglycolester derivatives and calcium nitrite-nitrate, which are non-chloride and non-alkaline, and was used for concreting at −5 °C. The results showed that with a w/c ratio of 0.45 and an increased admixture dosage of 5 L instead of 4 L per 100 kg cement, the compressive strength increased from 22.3 MPa to 30.1 MPa after 28 days of curing at −5 °C [34,35].

Demirboga et al. (2014) used a similar research set-up as Karagöl et al. (2013) but used a 6 wt% of urea, obtaining a workable mix with a maximum compressive strength of 16.38 MPa and 6.95 Mpa after 28 days of curing at −5 and −10 °C, respectively. An additional 28 days of water curing increased the strength to 30.28 MPa and 22.12 MPa, respectively [14]. However, dense microstructures were ensured when the curing temperatures were lowered down to −10 °C for specimens with antifreeze admixtures, by forming more C-S-H gel products and calcium hydroxide. Conversely, the control specimens showed more anhydrate particles and delayed the hydration process at −10 °C. Polat (2016) studied two groups of concrete mixes containing a 6 wt% of calcium nitrate or 6 wt% of urea for their impact on frost durability. Both mixes contained a constant amount of 0.5 wt% of super plasticizer and, directly after casting, the specimens underwent freeze–thaw testing in the temperature range of +/−10 °C. The concrete with the calcium nitrate admixture reached a 28-day compressive strength value of 28.05 MPa, while the urea-based concrete specimens developed a compressive strength of only 18.32 MPa [2].

Another study examined the effects of the combination of both admixtures of urea and calcium nitrate. An equal amount of 4.5 wt% was used in each concrete sample, and they were exposed to different curing regimes, i.e., −5 °C, −10 °C, −15 °C, −20 °C. The best results were obtained for the mixes cured at −5 °C and −10 °C, showing 41.91 MPa and 24.28 MPa, respectively, after 28 days. Concretes cured at −15 °C and −20 °C reached 8.86 MPa and 3.99 MPa, respectively [36]. There are mainly three aspects for strength development: 1) urea accelerates the hydration process by the rapid solubilization of C_3_S and C_2_S; 2) calcium nitrate, having similar ions to C_3_S and C_2_S nucleate and intensify the hydration process by rapidly forming crystallization products and 3) the eutectic point of urea and calcium nitrate ranges between −6.3 °C to −7.4 °C and −7.6 °C to −11.5 °C, respectively. Consequently, if the curing temperature is below the eutectic point, the compressive strength drops due to the ice formation in the pores and subsequently microcracks start to develop [20,36].

Wise et al. (1995) showed that the addition of a 3 wt% calcium thiocyanate admixture accelerated the strength development of concrete exposed to a temperature of −5 °C by 74%, when compared to the specimens cured at room temperature [27,45]. On the contrary, sodium thiocyanate non-chloride accelerator, an expensive admixture, enabled concreting at −7 °C, but over time it showed a high risk of an alkali–aggregate reaction (AAR), due to the high alkali content, and correspondingly released hazardous compounds categorized as Xn hazards [7,12,34,47].

Korhonen and Cortez (1991) evaluated the combinations of 6 wt% sodium nitrite + 2 wt% calcium nitrite and 6 wt% sodium nitrite + 0.06 wt% potassium carbonate. The test concretes performed well at temperatures down to −10 °C, reaching 35 MPa after 28 days [16]. However, studies reported that the pre-curing of these concrete specimens for 6 h at room temperature improves the mechanical properties, but further prolonging the curing at room temperature will develop microcracks due to the faster phase reaction between ettringites and monosulfates [57]. Mason and Schroder (1996) studied a mix containing sodium nitrite and calcium nitrite, which reached a compressive strength of 17.25 MPa when cured at −5 °C [39].

The addition of 3 wt% potassium carbonate antifreeze admixture to concrete, exposed to winter conditions, improved the strength by 30% by enhancing the hydration reaction of tri-calcium aluminate and tri-calcium silicate [44]. Conversely, specimens exposed to a 5 °C temperature produced more fiber-like products at an early age due to the faster hydration reaction which consequently led to the development of high internal stresses and microcracks forming a loose microstructure, thus reducing the efficiency of the admixture.

Concrete containing different combinations of antifreeze admixtures—such as 6 wt% sodium nitrite + 3 wt% sodium sulfate; 5% wt% calcium nitrate + 3 wt% sodium sulfate; 5 wt% sodium nitrite + 5 wt% calcium chloride; 5 wt% sodium nitrite + 5 wt% calcium chloride + 4 wt% of a superplasticizer—were produced and subsequently cured at −10 °C for 28 days. All the measured 28-day compressive strength values were lower in comparison to the reference concrete cured at 20 °C. However, curing at room temperature for another 28-day period boosted the strength and produced strength values comparable to the ones obtained for the reference concrete, as seen in Figure 2 [1,42]. Conversely, pre-curing the samples at room temperature for 24 h before exposing them to frost conditions will compensate for the slow hydration experienced during low temperatures. Studies using three different combinations of antifreeze admixtures, i.e., [NaNO_2_ + Na_2_SO_4_], [Ca(NO_3_)_2_ + Na_2_SO_4_] and [K_2_CO_3_ + lignosulphonate-based retarder] with dosages of 6 wt% + 3 wt%, 5 wt% + 3 wt% and 7 wt% + 0.75 wt%, respectively, and which pre-cured samples for 24 h at 20 °C followed by an exposure to a temperature of −10 °C, reached the strengths of 22 MPa, 22.4 MPa and 26.7 MPa after 28 days [43].

Concretes incorporating commercially available BASF non-chloride accelerators (14 L/m^3^), air-entraining agents (0.1925 kg/m^3^) and polycarboxylic-based superplasticizers (0.5775 kg/m^3^) cured for 28 days at 5 °C reached a compressive strength of 39 MPa. The chloride ion permeability measured after 56 days was limited to 190 coulombs when cured at 5 °C and to 150 coulombs for concrete cured at 20 °C, which is comparable and does not require a further increase in the curing temperature. As the length of the curing period increases, the formed ettringite gradually decreases and more C–S–H gel is formed leading to a dense microstructure [4,58].

Gagne et al. (1996) studied the effects of a naphthalene sulfonated-based superplasticizers when used in air-entrained concretes subjected to freeze–thaw cycles. Mixes containing over 1 wt% of the superplasticizer had a slump exceeding 200 mm. However, despite having a good slump and an air-void spacing factor of only 272 µm, the frost durability was significantly decreased. This is attributed to the adverse effect of the naphthalene-based superplasticizer, which decreases the hydration rate by reacting predominantly with C_3_A and adsorbing on the interstitial phase and free lime rather than on the calcium silicate phase. This reduces the surface hydration reaction and consequently increases the unreacted cement and porosity in the hardened paste [59,60]. Furthermore, concrete mixes with a w/c ratio of 0.43 and either containing 1–1.3 wt% of polycarboxylate ether or 1.8–2.2% of a poly-naphthalene sulfonate-based admixture cured at −25 °C reached a 7-day compressive strength value of 8.6 MPa, and 10 MPa after 28 days [61]. 

Ratajczak et al. (2019) studied the influence of diamidoamine salts on the compressive strength and freeze–thaw resistance of concrete. After 28 days of curing at room temperature, the control sample reached 72 MPa and samples with diamidoamine salt dosages of 0.10%, 0.25%, 0.50% and 1% had slightly lower values of 68.25 MPa, 64.42 MPa, 60.28 MPa and 59.73 MPa, respectively. However, after 50 days of frost action (F–T) the samples with diamidoamine salts had an increased strength of 0.42%, 0%, 1.54% and 8.64%, respectively, relative to the 28 days old samples. This is attributed to the formation of close spaced air bubbles within the binder matrix due to the incorporation of diamidoamine salts, consequently preventing frost damage. At the same time, the strength of the control specimen decreased by 58.8% [62]. 

Full-scale tests of concretes based on different combinations of chemical admixtures were demonstrated in Fort Wainwright, Alaska in 2008. The air temperature varied between 0 °C and −15 °C. The concrete contained 5–7 vol% of entrained air. The combination of a Glenium^®^3000—a high range water reducer, Pozzutec^®^20+—a non-chloride-based accelerator, Rheocrete^®^CNI—a corrosion inhibitor and Rheomac^®^VMA—a viscosity enhancer, was used. The casting area was divided into five sections (Figure 3a). Each section had a different dosage of admixtures, as seen in Table 2. The 28-day compressive strength of 48MPa was reached in Sections 1, 2, 3 and 5, and 38 MPa was achieved in Section 4 (Figure 3b) [10,63].

The formation of evenly dispersed air-voids by air-entraining agents (AEs), enhances the frost durability of concrete but at the same time tends to lower the compressive strength by crack formations gradually developing near the air-void system and at the paste–aggregate interface [25,64,65]. However, using up to 6% AEs will resist the frost action, but for every 1% increase above this limit the strength will decrease by 5% [66]. Franke et al. (2015) suggested a concrete mix with an admixture combination consisting of 4 wt% calcium nitrate + 0.16 wt% of an air-entraining agent (SIKA LPS A-94). The AE increased the porosity and the calcium nitrate supported the formation of fine air-voids which enhanced the frost durability. The strength development was accelerated, and the concretes also showed a low surface scaling of 82 g/m^2^ when subjected to 56 freeze–thaw cycles in a 3% NaCl solution. Concrete without admixtures showed surface scaling of 11,851 g/m^2^ [25,37,38].

The effects of various types of antifreeze admixtures on the freeze–thaw durability of concrete cast at −20 °C was studied by Grapp et al. (1975), as seen in Table 3. Mixes containing a potash admixture showed immediate internal damage. All concretes with other admixtures performed better in comparison with the reference mix (Figure 4) [1,67]. The formation of caustic alkalis was observed for mixes containing sodium nitrite and potash due to their reaction with Portland cement, thus excluding their application when reactive silica aggregates are used [1,28]. 

Concretes containing 1 wt% of an antifreeze admixture consisting of 30 wt% of calcium nitrate and 5 wt% of hydroxyethylamine reached a 28-day compressive strength between 28.42 MPa and 17.28 MPa while cured for 48 h at temperatures of 0 °C and −20 °C, respectively, followed by 26 days of water curing at room temperature [46,48]. Despite the favorable strength, the hardened concrete specimens exposed to corrosive environments of H_2_SO_4_ or NaCl with a 5% concentration for a period of 90 days at freezing temperatures showed detrimental effects. Relative to NaCl, H_2_SO_4_ was worse by seemingly dissolving the concrete’s free lime and increasing the pores’ space volume, water absorption and consequently decreasing the strength properties irrespective of the curing temperature [48]. Furthermore, any corrosive chemicals or salts over a 3% concentration will lead to the deterioration of the hardened concrete surface and affect the frost durability (F–T cycle) [68,69].

A summary of the 28-day compressive strength values obtained for OPC-based concretes incorporating various types of antifreeze admixtures is shown in Figure 5. The urea- and nitrite-nitrate-based admixtures provided acceptable strength values while at curing temperatures of down to −10 °C. Concretes containing 3 wt% of calcium chloride reached 32 MPa at −10 °C. However, higher amounts of this admixture decreased the compressive strength. OPCs being exposed to −20 °C for short period followed by curing at a normal temperature accelerated the strength development. Based on the collected data, the lowest curing temperature tested for most chemical admixtures used in OPC was −10 °C. Few tests were done at even lower temperatures.

## 3. Composite Portland Cement-Based Concretes

Composite Portland cements are a mixture of OPCs and SCMs or other types of ecological cements. These composites are often used to reduce the energy consumption of OPC production, to decrease greenhouse gases and to efficiently utilize SCMs coming from industries as byproducts (slag, fly ash, silica fume, etc.). Moreover, partially replacing OPCs with SCMs has the ability to enhance the properties of concrete, depending on the type and amount of SCM. Despite having a slower hydration rate compared to pure OPC-based concretes, composite Portland cements a gain more or less similar strength but at later ages [70]. This is due to the reaction with Ca(OH)_2_ from cement hydrates. Consequently, this leads to the production of a denser structure and also improves the impermeability and sulfate resistance when cured at an ambient temperature [71,72]. However, these cements will behave differently at low curing temperatures and for different chemical admixtures. This section will briefly describe the effects of chemical admixtures on the mechanical and frost durable properties of composite Portland cement at low curing temperatures.

Jiang et al. (2013) studied the strength and freeze–thaw durability of composite concrete mixes containing a 6 wt% of fly ash and 4 wt% of “LNC-53” non-chloride-based antifreeze admixtures along with a naphthalene-based water reducer and 5.3 wt% of an air-entraining agent. The test concretes had different curing regimes, i.e., pre-cured at an ambient temperature for 0, 4, 12, 24, 48 and 72 h, followed by 28 days of curing at −10 °C. The specimens reached a 28-day strength between 12.6 MPa for 0 h and 38.6 MPa for 72 h of pre-curing, while the reference specimen developed a strength of 68.3 MPa having been cured at an ambient temperature. However, 72 h of pre-curing followed by 7 days of being exposed to a temperature of −10 °C and an additional 28-day standard curing compensated for the negative effects of the low temperature and the samples reached a final strength of 66 MPa. Furthermore, specimens pre-cured for 48 h and 72 h at ambient temperatures formed a dense binder matrix by promoting the acceleration effect with sharp increases in the pozzolanic activity [70], consequently showing a minimal weight loss of only ≤ 1% and a relative dynamic modulus of elastic (RDME) loss of 15% after 300 cycles. Conversely, a shorter period of pre-curing (0 h to 24 h) significantly slowed down the formation of a dense matrix and consequently led to a high percentage of weight loss (>5%) and a drop in the RDME of over 40% before completing 175 F–T cycles [73]. Following the experimental program from Jiang et al. (2013), Dong et al. (2013) increased the cement substitution by fly ash to 10 wt% and 20 wt%. The results conclude that increasing the fly ash content causes the strength to decrease regardless of the curing conditions due to fewer hydration products [74]. Moreover, at lower curing temperatures (−10 °C), the hydration rate is decreased. This is visible in the measured 28-day strength values of 35 MPa for 10 wt% fly ash and 25 MPa for 20 wt% fly ash replacement. Despite having relative low strength values after 28 days, the strengths increased linearly and reached 48 MPa and 40 MPa at 90 days. These values are quite close to the 28-day strength of standard cured specimens [73,75].

After showing that a 20 wt% fly ash replacement has the potential to achieve a satisfactory strength despite the low temperature curing, Jiang et al. (2015) evaluated the frost durability of the hardened concrete mix exposed to corrosive (5% sodium and 5% magnesium) sulfate solutions. The mix is blended with 20 wt% fly ash, a 0.5 wt% naphthalene-based superplasticizer and a 4.2 wt% SJ-3 air-entraining agent. The used air-entraining agent (SJ-3) is a saponin-based surfactant. Saponin is found in Gleditsia Sinensis plants, is soluble in water and is stable in acidic and alkaline environments. After exposure to 400 freeze–thaw cycles in the sodium sulfate or the magnesium sulfate solutions (Figure 6a), the concrete exposed to the magnesium sulfate solution showed a relatively higher corrosion and strength loss than the concrete exposed to the sodium sulfate solution. This is attributed to the replacement of calcium ions from C–S–H by magnesium ions, forming M–S–H (magnesium silicate hydrate). Furthermore, the sulfate solution led to the formation of more ettringite and gypsum (Figure 6b) and consequently developed more microcracks and expansions [76,77,78,79]. However, the corrosion is less intensive due to the used air-entraining agent which forms more and smaller air bubbles, and reduces the surface tension at the interface due to the hydrophilic and hydrophobic ends of the molecules [80].

With no cold weather admixtures (CWAs), and by further increasing the fly ash replacements up to 30% and using a 0.9% polycarboxylate superplasticizer, the samples gained a strength of 68.1 MPa after 90 days of standard curing. However, after exposure to 100 F–T cycles, the strength decreased by 84%, due to the non-inclusion of cold weather admixtures, leading to the termination of the pozzolanic reaction of fly ash [81]. Concretes containing between 10 wt% and 15 wt% of the fly ash showed a good frost durability up to 300 F–T cycles. The measured relative dynamic modulus of elastic (RDME) was between 87.6% and 82.5%. This result was related to a large surface area of the fly ash, which densified that pore structure. On the contrary, increasing the fly ash content beyond 15% decreased the frost durability mainly due to increased air-void spacing [82].

Moreover, instead of replacing cement, fly ash-substituting aggregates at 20, 40, 60 wt% will pronounce the compactness of the mix by filling the voids and showing a durability factor over 70% and strength loss of less than 20% after 300 F–T cycles [83]. Furthermore, CO_2_ curing significantly decreased the surface scaling of the OPC and 20 wt% fly ash replaced concrete after 150 F–T cycles (Figure 7). This is attributed to the fact that CO_2_ curing decreases permeability and consequently enhances strength and leaves less pore space for ice crystallization during frost action [84,85,86].

Ogurtsova et al. (2017) studied a composite binder blended with finely ground feldspar, sand, a melamine-formaldehyde-based plasticizer and commercially available antifreeze admixture—either MC Rapid 025 or MC Rapid 015—at a temperature of −20 °C. The mixes with either 50 wt% or 30 wt% replacement by ground feldspar and sand reached strength values of 32.16–34.05 MPa and 46.48–46.69 MPa for 4% to 5% of admixture after 28 days. The strength gain is mainly attributed to the fusion between the admixture and cement, resulting in the dissolution of silica compounds from cement and consequently forming salt structures and maintaining the liquid phases in freezing temperatures [9,17]. Sadowski et al. (2020) studied air-entrained concretes containing between 10 wt% and 30 wt% of either quartz–feldspar or basalt mineral powders, cured at standard conditions. The measured 28-day compressive strength decreased from 37.55 MPa to 29.73 MPa when 1.1% of an AE was used. Mixes containing mineral powders showed a decrease in their 28-day compressive strength to 8 MPa. The used mineral powders decreased the diameters of formed air-voids from 1 mm down to 0.130–0.5 mm. The authors concluded that despite a lower strength the test concrete had a better frost durability due to a lower air-void spacing distance [87]. 

Another type of composite concrete mix incorporating condensed silica fume (0, 20, and 30 wt%), a sulfonated naphthalene formaldehyde (SNF) superplasticizer and resin-based air-entertaining agent was studied at a curing temperature of 10 °C by varying the w/b ratio between 0.25 and 0.55. Irrespective of the amount of cement replaced by silica fume, the air-entrained concrete, with a w/b ratio of 0.55, had a lower strength relative to the non-air-entrained specimen, which had a w/b of 0.25. This lower strength is attributed to the combined effect of a high w/b ratio and air-entrainer which generates a high porosity within the binder matrix despite the silica fume [88,89]. However, due to the silica fume, the coarser pore size is decreased, and so more and smaller pores retain the same pore volume [90]. The frost durability of the silica fume composite concrete can be enhanced either by lowering the w/b ratio or by reducing the volume of the capillary pores, consequently leading to less scaling [23,91,92,93,94]. The presence of the amorphous silica fume altered the frost damage mechanism and acted as a strength accelerator due to the formation of a denser microstructure having a lower water penetration during freezing. Therefore, even without the presence of the air-entrainment a good frost durability is achieved (Figure 8) [89,95,96,97].

Based on previous studies, using over 15% silica fume corresponds to a high-water requirement. Therefore, 15 wt% silica fume is considered as the optimal amount for cement replacement, reaching a strength of 65 MPa with no air-entraining agent and 45 MPa with a 0.15% air-entraining agent for standard curing [91]. The reduction in strength is due to the carbon originating from the silica fume. The carbon gradually adsorbed the air-entraining admixture and prevented the formation of a stable and uniformly spaced air-void system, which consequently lowered the compressive strength [94]. Conversely, the surface scaling after 28 F–T cycles is not in line with the obtained strength showing 425 g/m^2^ and 375 g/m^2^ for the reference and air-entrained concrete, respectively. This is attributed to the relatively higher moisture uptake of 2.25 Δwn for the reference concrete, compared to the 1.25 Δwn for the air-entrained silica fume concrete [91].

Jang et al. (2015) studied the strength properties of high-performance concretes (HPCs) with the fixed amount of silica fume of 5% and following curing temperatures of 5, −5 and −15 °C. The strength values gradually decreased with sinking temperatures, reaching 25.41 MPa, 23.76 MPa and 14.02 MPa after 28 days regardless of the addition of a 0.02% of air-entraining agent [98]. However, standard cured HPC containing up to 7 wt% of silica fume and a w/b ratio of 0.3 exposed to 56 F–T cycles in a 3% NaCl solution showed very limited surface scaling of less than 500 g/m^2^ and a drop in the RDME to around 90%. This is because of the dense microstructure showing a transition zone <5 µm with lower Ca/Si in the ITZ, due to the pozzolanic reaction of silica fume with Ca(OH)_2_ to produce more C–S–H gel. However, a high Ca/Si ratio in the bulk paste was observed because of the fine dispersion of portlandite and C–S–H gel. However, by increasing the w/c ratio, the ITZ width increased in the range of 30–40 µm and the porosity also increased which consequently amplified the uptake of the pore solution (NaCl) during F–T cycles and delivered a detrimental effect by forming cracks (Figure 9) [50,99,100].

Furthermore, reducing the w/b ratio to 0.22 and varying the silica fume (0–25 wt%) in the sulfate resisting cement (Type V), the frost durability and chloride penetration after 60 F–T cycles showed a mass loss (surface scaling) of 42.2 g/m^2^ for specimens containing no silica fume and, with 25 wt% silica fume, a mass loss of 18.2 g/m^2^ was registered. This is again due to the formation of a dense microstructure. However, the chloride permeability had contradictory results, showing 57 coulombs for 0% of silica fume and 939.5 coulombs for 25% silica fume. However, in general the chloride penetration lies within the negligible (<100) and very low (100–1000) range [101]. Similar results were obtained back in 1993 in tests on the sulfate resistance of concrete, delivering a durability factor of over 90% after 900 F–T cycles for up to 20 wt% silica fume replacement, while the reference mix dropped to 60% after 58 cycles [102]. Likewise, Mardani-Aghabaglou, et al. (2014) evaluated the sulfate resistance (sodium and magnesium sulfate) of concretes containing 10 wt% of silica fume which were immersed for 300 days. The results concluded that the formation of ball ettringite, which is by its nature non-expansive unlike needle ettringite (reference concrete), reduced the formation of cracks (Figure 10) [103,104]. It showed a weight loss of only 0.48% and a reduction in strength of 7% after being subjected to 300 freeze–thaw cycles [104]. The positive effect was related to the formation of a micro air-void layer around the fine aggregate particles of the silica fume. Another reason for this is that, by replacing cement by mineral admixtures, the C_3_A content is reduced resulting in less ettringite being formed and a further increase in the consumption of the Ca(OH)_2_ due to the pozzolanic activity, leading to a denser binder matrix [89,90,104].

Ground granulated blast furnace slag (GGBS), another SCM byproduct from steel industries, is widely accepted when compared to fly ash, as the amorphous silica in GGBS reacts with the Ca(OH)_2_ relatively faster at an early age producing more C–S–H gel. However, at low temperatures, it has a negative impact on the hydration rate and therefore on the properties of concrete. Studies conducted on the standard cured 50 wt% replaced slag self-compacting composite concrete, showed an RDME of up to 80% after continuous 180 F–T cycles mixed with no entrained air [101,105]. Further increasing the replacement level showed an extensive internal damage with an RDME below 20% after 180 F–T cycles. This is due to the formation of ice in the capillary pores during the F–T action, which induces expansion stress on the wall of the pores and promotes crack formations. However, this ice–stress mechanism can be alleviated either by reducing the volume of the capillary pores or by inducing entrained air to release the expansion stress [106,107,108]. In contrast, concrete containing 5% silica fume along with 50% GGBS filled most of the capillary pores by forming a dense microstructure, which revealed an RDME of 95%. Keeping the GGBS constant at 50%, and increasing silica fume up to 15%, eventually reduced the cement content and hydration products and therefore exhibited a lower F–T resistance (RDME—65%) after 180 cycles [106].

Similar experiments conducted on UHPC by a partial replacement of cement with slag (0 wt% to 65 wt%) and an additional 25 wt% of silica fume, showed a maximum weight loss of 200 g/m^2^ and an RDME between 95% and 100% after 100 F–T cycles. This is mainly attributed to the low w/b ratio of 0.22 and densified microstructure, due to the pozzolanic effect from silica fume which consequently reduces the volume of freezable water because of the finer pores and reduced uptake of moisture (frost suction) [105,109]. However, adding mineral admixtures rich in silica will lead to a reaction with the cement hydration product Ca(OH)_2_, which eventually forms a more stable ettringite (C_6_AsH_32_) and siliceous hydrogarnet (C_3_AS_0.8_H_4.4_), having a lower Gibbs free energy and ultimately being a more stable product than monosulfoaluminate [110].

An interesting full-scale casting of a composite concrete containing 23 wt% of fly ash was done while casting a concrete floor in the operational freezer of a dairy food manufacturing plant. The concrete work was done at −4 °C. The used composite concrete mix measured an initial temperature of 27 °C while casting. It contained a sodium thiocyanate-based antifreeze admixture (3910 mL/100 kg) and reached a strength of 27.6 MPa after 28 days of curing [7]. Another example was the manufacturing of pile caps for a pier in Gloucester, Massachusetts. The used composite concrete contained 70% type II cement, 30% blast furnace slag and sodium thiocyanate as an antifreeze admixture. The concrete was cast at −3 °C and reached a 28-day compressive strength of 49 MPa [7].

Sodium nitrite, a well-known antifreeze admixture in the field of OPC-based cold weather concreting, can be compared with tertiary blended concrete in terms of its strength development at low temperatures. A steel fiber reinforced composite concrete containing a combination of 12.5 wt% fly ash and 5.4 wt% silica fume, blended with a 0.4 wt% of sodium nitrite antifreeze (AF) admixture and a 1.8 wt% of polycarboxylate superplasticizer water-reducing agent, gained a strength of 73.8 MPa after 28 days with a curing regime of −5 °C for 7 days followed by 21 days of standard curing. However, increasing or decreasing the AF to 0.6% or 0.2% decreased the strength development. Less hydration products were registered for lower dosages of AF, while a higher dosage (0.6%) revealed a dense microstructure of C–S–H gel and large ettringite needles in pores. The needles overlapped and expanded and consequently released the stresses by forming microcracks [40,111]. Moreover, a similar type of blended concrete, with no AF or AE admixtures, deteriorated linearly with every F–T cycle and decreased the RDME to 15% after 210 F–T cycles. This is attributed to the high w/b ratio and the non-intrusion of the air-entraining agent, which limited the air-void formation to release the stresses formed by the ice. Consequently, these two aspects affected the formation of cracks internally during the F–T cycles [112,113]. In contrast, Yazici (2008) studied the frost and frost deicing salts (NaCl) durability of hardened composite concrete specimens containing 30–60 wt% of class C fly ash, 10 wt% of silica fume and which had a low w/b ratio of 0.28. All specimens reached a strength of around 70 MPa after 90 days. The measured residual compressive strength increased to 108% for up to 30% fly ash replacement, showing that during the F–T action the Na^+^ ions acted as additional curing media and the residual splitting tensile strength dropped only by 1% or 2% after 90 freeze–thaw cycles. However, a further increase in the amount of fly ash decreased the strength development and frost durability despite the added silica fume. This negative impact is due to the high amount of unreacted fly ash (Figure 11) [114].

By introducing an air-entraining agent to a ternary mix containing 15–35 wt% of fly ash and 5 wt% silica fume, an RDME and durability factor value over 80%, with very slight to moderate surface scaling after 300 F–T cycles, was obtained. These positive effects are mainly credited to the improved air-void system, which created the micro and macro air-size voids that resist the internal stress. However, increasing the amount of fly ash will eventually increase the unburnt carbon and adsorb the air-entraining agent, consequently forming an unstable air-void system by eliminating smaller air-voids and converting them to bigger air-voids with a higher spacing factor [82,94,115,116].

However, a novel air-entraining (AE) agent, the polyethylhydrosiloxane (PEHSO) admixture has very high potential to resist the frost action. It is siloxane-based and highly reactive with the hydroxyl group of the hydration product, reacting with CH to liberate hydrogen and consequently produce evenly distributed water repellant micro air-voids [117]. Sobolev and Batrakov (2007) studied the frost durability of hardened composite concretes containing 15 wt% of silica fume with either 15 wt% ponded ash or 45 wt% finely ground blast furnace slag. All mixes contained a 1.8% SNF-type superplasticizer and 0.0625% PEHSO air-entraining admixture. The produced composite concretes had an air-void spacing of less than 270 µm, with an average size between 30 and 70 µm. They resisted the F–T action over 700 cycles with only a minor deformation less than 0.05% [26]. Increasing the pond ash to over 15% showed a detrimental deformation effect before 200 F–T cycles. Due to the relatively slow reaction of pond ash, a major portion of the hydration product (CH) is utilized by the AE, leaving behind partially unreacted ash particles [118].

A new alternative for partial cement replacement, CKD (cement kiln dust), a byproduct from cement manufacturing, is widely utilized in construction industries [119]. Replacing cement by 15 wt% of CKD increases the strength by 7.5% compared to the reference concrete. This is attributed to the dissolution of silicates consequently forming C–S–H and high amounts of free lime in the CKD reacted with water at a faster rate by producing CH, resulting in an increased water demand to maintain the acceptable consistency due to the high particle fineness. However, further increasing the CKD will substantially reduce the strength as it has very low hydraulic properties and increases the porosity and microcracks [84,120,121,122]. This indicates that samples containing up to 15% CKD replacement resisted the 300 F–T cycles with a durability factor of over 70% and a dynamic modulus of elasticity of 55 GPa, because of the low porosity volume and low water absorption compared to a higher CKD replacement [122].

Another approach to accelerate the hydration is to introduce “mechano-chemical activated binders”, for instance, a finely ground slag and recycled white cement mixture with OPC, elevates the maximum hydration rate and strength development in a shorter period of time and is therefore subsequently used in winter construction [123]. This is attributed to the activated slag enhancing the surface area and the alkaline medium by the presence of silicate ions leading to the promotion of the hydration process. So far, as a new topic in the research field, very little research have been done concerning the behavior and properties under winter conditions.

## 4. Alkali-Activated Concretes

The amount of data related to concretes based on alkali-activated binders regarding their performance in cold conditions is generally very limited. Only few data related to their properties when cast at low and very low temperatures were found.

An alkali-activated system (AAS) is an activation of vitreous structures fully or partially using chemical processes to convert itself into a cementitious skeleton [124]. The alkali-activated material precursors are highly rich in aluminosilicates—for instance, fly ash, silica fume or slag. For the activation of these systems, alkaline chemicals such as sodium silicate, sodium hydroxide, potassium hydroxide, sodium carbonate, sodium sulfate or combinations thereof are used [125]. The AAS has been developed for many decades and many researchers have found that systems activated with sodium silicate have superior strength and durable properties in optimized alkaline solutions, with a preferable lower modulus and work conditions [124,126,127,128]. Generally, mixes with a Si/Al ratio between 3.0–3.8 and a lower Ca/Si ratio achieve a high strength and more durable properties [129,130,131]. However, the curing temperature plays a notable role on the strength properties of either alkali-activated or OPC systems. At low curing temperatures or during winter, construction activities are delayed due to retarding effects, which significantly increase the construction costs [132]. This section will briefly describe the effects of the curing temperature and/or available chemical admixtures on the properties of alkali-activated concretes.

A few alkali-activated systems have poor workability affecting the construction work. The rheology can be improved by using either a lignosulphonate admixture, enhancing the workability due to its non-polar molecules and low coulomb attraction, or using an air-entraining agent which eventually develops air-pores and subsequently improves the workability [126,127]. These AASs consume a higher dosage of air-entraining agents compared to the OPC system. During the F–T cycle, the air-entrained blast furnace slag concrete mix activated with sodium silicate shows relatively less internal damage measuring a durability factor of over 65%, while sodium carbonate-activated mixes exaggerated the damage with a durability factor of only 9% after 300 F–T cycles [126]. This is mainly attributed to the uniformly distributed fine air-voids formed within the AAS, while the sodium carbonate had a random distribution of air-voids, consequently affecting the frost durability. However, lower amounts of the sodium silicate increased the internal damage, which was related to the incompatibility between the air-entraining agents and the alkali-activators [133].

Ya-min et al. (2015), studied the microstructure of the alkali-activated slag pastes at different temperatures (7, 15, 20, and 30 °C) using a 6% water glass activator with a modulus value of 1.42. Similar to the OPC paste, with an increasing temperature the alkali-activated paste shortened the initial and final setting time and enhanced the compressive strength development [128,134,135]. The flexural strength increased until 15 °C. Further increases resulted in an adverse effect. This reduction was attributed to a higher chemical shrinkage and microcrack formation, as seen in Figure 12. In this case, a faster formation of C–S–H, C–A–S–H and N–A–S–H occurred. It can be concluded that an alkali-activated system, which is prone to high shrinkage at high temperatures, can be mitigated by low temperature curing with acceptable strength properties and a reduced development of microcracks [128].

Alkali-activated slag concrete was used in a full-scale application at 5 °C at the CRICS office building in Chongqing Jianke, China (Yang et al. (2018) [136]). The 28-day compressive strength reached between 40–60 MPa. The control samples kept at 20 °C and RH > 95% showed a 28-day compressive strength between 60–80 MPa [136].

Besides, Zhang et al. (2020) suggested that, rather than using pure alkali-activated slag concrete, the usage of a composite alkali-activated GGBS/PC for winter constructions is more beneficial as it combines the hydration from cement reactions and the polymerization from alkali-activated slag [6]. The selected concrete had 100% OPC, 100% GGBS and 30, 20, 10 wt% slag replaced by cement and cured at temperatures of 20 °C, −5 °C, and −20 °C. The results showed that specimens containing alkali-activated GGBS/PC had higher strength values compared to pure alkali-activated slag concrete or OPC, regardless of the curing temperature (Figure 13). The Na+ and OH− ion concentration of the alkali-activated solution is 20 (Na^+^) and 3 (OH^−^) times higher than OPC and allowed the reaction to continue despite the lower temperature. Consequently, a dense microstructure is formed with continued hydration and a faster conversion of sulfate to sulfoaluminates. Additionally, the alkali solution has a lower freezing point and can act simultaneously as an accelerator to OPC by producing more C–S–H [6,137,138,139].

Adding a nano-material like 3 wt% of nano-silica (NS) increases the frost resistance of alkali-activated slag concrete [54,135,140]. The strength measured after 300 freeze–thaw cycles decreased by 6.14% for the reference mix and 5.56%, 4.89%, and 4.1% for the alkali-activated slag concretes incorporating 1 wt%, 2 wt% and 3 wt% of NS, respectively (Figure 14) [54]. The addition of NS enhanced the durability by more C–S–H gel formation and the denser microstructure led to a decrease in the water permeability leading to a low saturation degree during frost action [54,130].

The early age and ultimate compressive strength of metakaolin-based geopolymer concrete (GPC), activated by using sodium silicate subjected to curing temperatures between 10 °C to 80 °C, was studied by Rovnaník (2010). Lower temperatures delayed the strength development, but the ultimate strength values were higher or similar to mixes cured at higher temperatures (62 MPa (Figure 15a)) [129]. At high temperatures, most of the geopolymerization reaction products appears at an early age but fail to prolong until a later age and further generate microcracks. Additionally, due to the faster reaction, the phases change from amorphous to crystalline rapidly and also lose water, resulting in a high porosity and voids [141]. At low curing temperatures, the reaction takes place slowly by refining the pore structure, and consequently a lower porosity and a more compact structure is obtained. No crystalline phases were depicted. Figure 15b depicts the bulk density of concrete after 28 days of curing at the respective temperature [129,142].

The F–T durability of the hardened alkali-activated concrete, containing 50 wt% waste ceramic powder (WSP) rich in SiO_2_ and 50 wt% slag (GGBS) rich in CaO, was studied. Additionally, another mix of 50 wt% WSP and replacing 0–40 wt% of slag with fly ash rich in Al_2_O_3_ was investigated. All the specimens were activated with sodium silicate with a modulus value of 0.75 and reached a compressive strength of 72.1 MPa for 50 wt% WSP and 50 wt% GGBS after 28 days of standard curing, and 45.9 MPa for the 40 wt% GGBS replaced with fly ash mix. It was observed that the residual strength and residual weight linearly decreased with an increasing fly ash level for every F–T cycle (Figure 16a,b) [143]. The CaO, which is main constituent of GGBS, enhanced the formation of C–A–S–H, but with the increasing fly ash eventually the CaO content decreased and restricted the C–A–S–H gel formation. Consequently, the unreacted silicate and more uneven void formations within the binder matrix influenced the growth of ice crystals [143,144].

Alkali-activated fly ash concrete mixes with and without an air-entraining agent showed a strength loss of 5% for non-air-entrained alkali-activated concrete, while the samples containing a 0.2 wt% air-entraining agent (MB-VR-Master Builders) showed no strength loss after 300 freeze–thaw cycles. In contrast, the equivalent OPC with and without an air-entraining agent showed a strength loss of 5% and 20%, respectively (Figure 17a) [145]. Similarly, the reduction in the dynamic modulus of elasticity was 6.8% and 8.4% for mixes with and without the air-entraining agent, respectively, for fly ash-based alkali-activated concrete (Figure 17b) [145,146].

Alkali-activated concrete based on a mixture of 65% of blast furnace slag and 35% of fly ash activated with a liquid sodium silicate reached a compressive strength of 65 MPa and showed a good frost durability when exposed to 80 freeze–thaw cycles [147]. Others observed that, after undergoing 50 freeze–thaw cycles, these concretes had an improved compressive strength. The addition of polypropylene fibers enhanced the frost durability of such concretes further [148,149]. Alkali-activated concrete containing 50 wt% of GGBS and 50 wt% of fly ash activated with an 8 M NaOH solution with 0.5 wt% of polypropylene fibers, showed, after 50 freeze–thaw cycles, an increase in the compressive strength by 11.2%, compared to the value obtained before freeze–thaw cycling. The strength increase was related to the slow reaction of the slag and formation of more alkaline products during the thawing phase of the F–T cycles [148].

The frost durability of three different alkali-activated slag concretes activated with various alkaline solutions—(N/C)—(NaOH/Na_2_CO_3_); (WG)—(Water glass); (N/C-25)—((NaOH/Na_2_CO_3_) + (25% water glass))—was compared with OPC. The slag concrete activated with water glass retained 60% of the RDME and had a mass loss of about −1.0% while subjected to 300 cycles of freeze–thaw [150]. The OPC reference mix and the mix with (N/C) and (N/C+25) showed an RDME value of under 60% and a mass loss of −1.0% before 75 freeze–thaw cycles (Figure 18) [150].

## 5. Discussion

The efficiency of the antifreeze admixtures differs significantly depending on the used binder system. In general, the ultimate 28-day compressive strength tends to decrease with a lower curing temperature. The effectiveness of the AF admixture in limiting this trend depends on the binder system. In the case of OPC-based concretes, the maximum strength was achieved from a combination of urea and calcium nitrate when cured at −5 °C (Figure 19). A decrease in the temperature to −10 °C lowered the 28-day strength by almost 50%. This can by related to the eutectic temperature point (−10 °C) of the used chemicals. At temperatures below −10 °C, the strength deteriorated dramatically and reached the highest value of 5 MPa at −20 °C. Significantly, lower values were measured for concretes containing calcium nitrate. In this case, the compressive strength dropped from 33 MPa, when cured at −5 °C, down to 10 MPa and 5 MPa when cured at −10 °C and −15 °C. Thus, in general, based on the obtained literature data, the lowest curing temperature—which provides somehow acceptable results over 28 days—for OPC is between −5 and −10 °C. Lower temperatures result in a very low strength independently of the used AF admixture. 

Surprisingly, the performance of some of the composite cements and alkali-activated binders at low temperatures appeared to be better in comparison to OPC. A similar trend of decreasing 28-day compressive strength values with lower temperatures was observed (Figure 20). However, the ultimate strength at curing temperatures below −10 was higher in certain cases. Most concretes based on composite cements should be pre- or post-cured at an ambient temperature. At the same time, some combinations enable the development of a high strength when cured even at −20 °C. One example is a mix based on a combination of 30% of feldspar and 70% of OPC and containing 4% of the MC-Rapid (AF). In this case, the 28 days compressive strength reached 46 MPa without the need of any pre- or post-curing. The dissolution of silica compounds and generation of a high amount of C–S–H in a short period was indicated as the main reason for this result.

The alkali-activated binders that use strong alkali activators generally developed the highest strength values when cured at −15 and −20 °C (Figure 20). The alkaline solution acts as an accelerator and activator, but it also lowers the freezing temperature of the pore solution. This enables a continuation of strong geopolymerizations and hydration process even at extremely low temperatures. One example is a concrete containing 80% of blast-furnace slag (BFS), 20% of OPC and activated with sodium silicate. The compressive strength reached nearly 40 MPa when cured continuously at −20 °C.

The efficiency of the AF admixtures in OPC and composite-based cement is summarized in Table 4. Only the nitrate-based AF admixture, appeared to be equally efficient for OPC and composite cements. The urea-based AF admixture seems to be also more efficient when combined with calcium nitrate. Additionally, it tends to enhance workability and to accelerate strength development.

The study showed that most of the available admixtures are equally efficient for concretes based on OPC and on composite cements (Table 5). The ultimate effect depends on the type of the used SCM materials. For example, in the case of concretes containing a mixture of OPC with SF the resin-based AE appeared to be inefficient. It tended to generate a high porosity, which lowered the frost durability.

Concretes based on alkali-activated binders use strong alkaline solutions, which act as activators, accelerators and AF admixtures at the same time. The freezing point of the pore solution is significantly lowered, thus enabling hydration and geopolymerization even at subfreezing temperatures. The AE admixtures tend to be deactivated by some of the alkali activators. For example, sodium carbonate produces unstable air-voids, while, in contrast, a fine distribution of air-voids is achieved with the sodium silicate activator. Further, the amount of fly ash should be limited in composite alkali-activated systems, due to the slow reaction of fly ash (Table 6).

## 6. Conclusions

Casting of concrete in cold weather conditions is generally a complicated and costly process and it can be problematic to achieve all the required short- and long-term performances. Fresh concretes exposed to low temperatures show delayed setting times and get slower if there is any strength development, an increased risk of freezing, etc. Two groups of methods are used to prevent those problems. The first includes the usage of heating systems, protective covers or enclosures. The second group is based on the usage of suitable materials, additives, compositions, etc.

The preformed literature review showed that the present knowledge enables the production of concretes based on OPCs that can be used at temperatures as low as −10 °C without the need for the application of any additional measures. The developed hydration heat combined with chemical admixtures are enough to develop an adequate strength as well as freeze–thaw durability.

At the same time, this study revealed a significant lack of knowledge for concretes in which Portland cement is partially or fully replaced by SCMs. The SCM blended mixes are either pre-cured at an ambient temperature or are shortly exposed to a lower temperature. Apparently, this is not the case in extreme winter conditions, which provides a low temperature all the time. On other hand, concretes based on binders with SCMs exposed to lower temperatures showed problems related to the fresh concrete properties and the long-term durability. The lack of knowledge is even more evident in alkali-activated systems (AASs), especially regarding the early age strength development and microstructures at very low temperatures. At the same time, the few published results showed that these binder systems (AAS) perform the best at extremely low temperatures and are able to gain up to 40 MPa and more when cured at −20 °C. Certainly there is urgent need to perform basic studies as well as a full-scale test to verify the effects of cold weather and arctic conditions on the early and long-term performances of ecological (SCMs and AAS) concretes.

## Figures and Tables

**Figure 1 materials-13-03467-f001:**
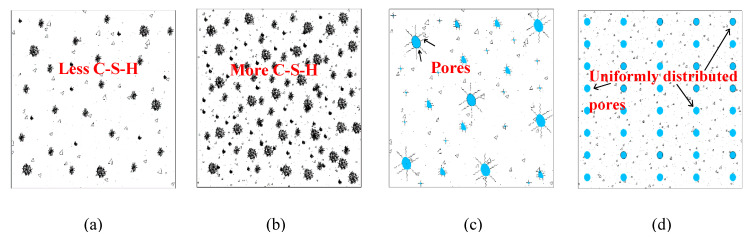
Low temperature cured/casted concrete (**a**) without antifreeze admixture; (**b**) with antifreeze admixtures; (**c**) without an air-entraining agent; (**d**) with an air-entraining agent.

**Figure 2 materials-13-03467-f002:**
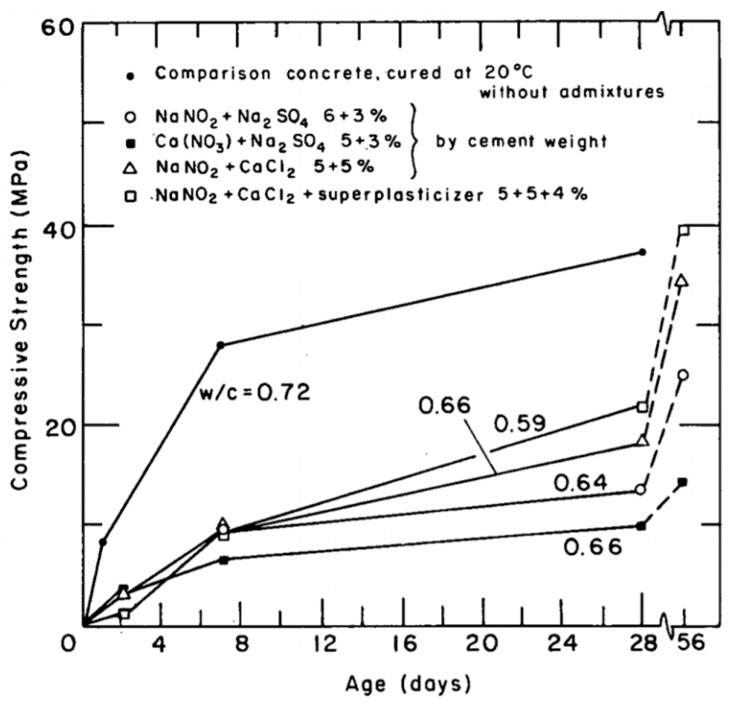
Compressive strength of ordinary Portland cement (OPC) concrete cured at 20 °C vs. antifreeze concrete at −10 °C. Dashed line corresponds to room temperature curing (reprinted from Kivekäs et al. 1985, © VTT Technical Research Centre of Finland Publishers).

**Figure 3 materials-13-03467-f003:**
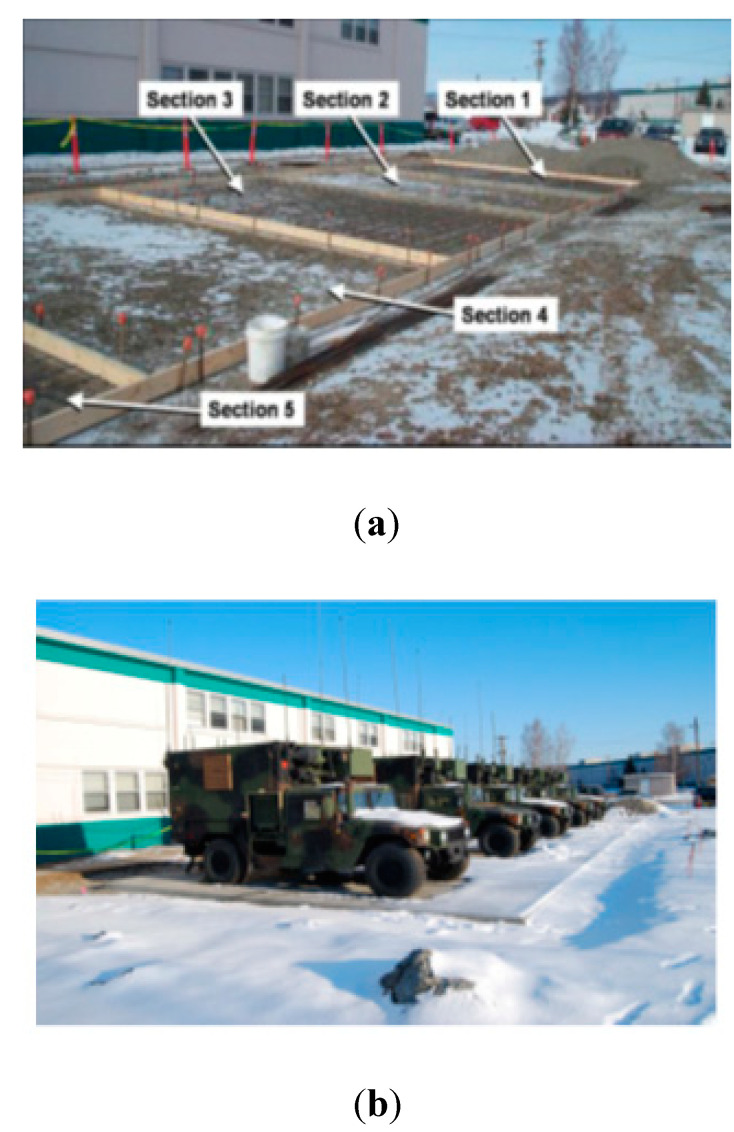
Concrete hardstand for a military vehicle (**a**) before and (**b**) after casting, Fort Wainwright, Alaska (April 2008 (reprinted from Barna et al. 2010, © U. S. Army Engineering Research and Development Centre, Cold Regions Research and Engineering Laboratory Publishers)).

**Figure 4 materials-13-03467-f004:**
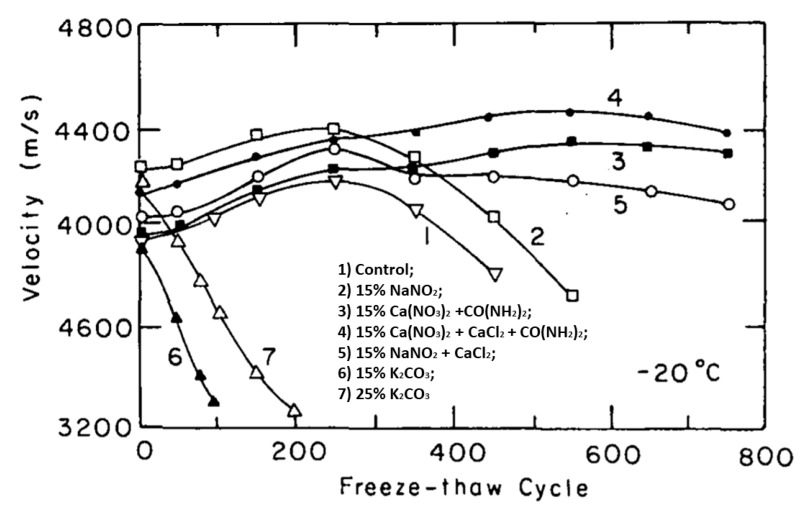
Freeze–thaw durability of concrete with the admixture concentration (reprinted from Korhonen 1990, © U. S. Army Corps of Engineers, Cold Regions Research and Engineering Laboratory Publishers).

**Figure 5 materials-13-03467-f005:**
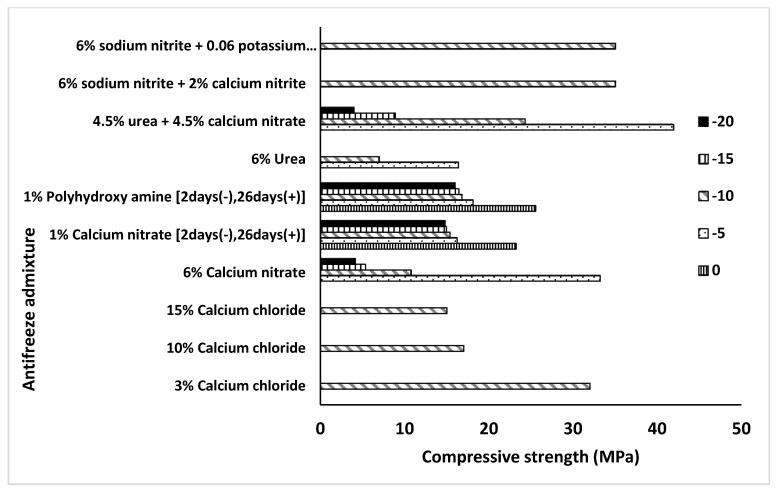
Effects of various types of chemical admixtures on the 28-day compressive strength of concretes based on OPC cured at 0, −5, −10, −15, and −20 °C.

**Figure 6 materials-13-03467-f006:**
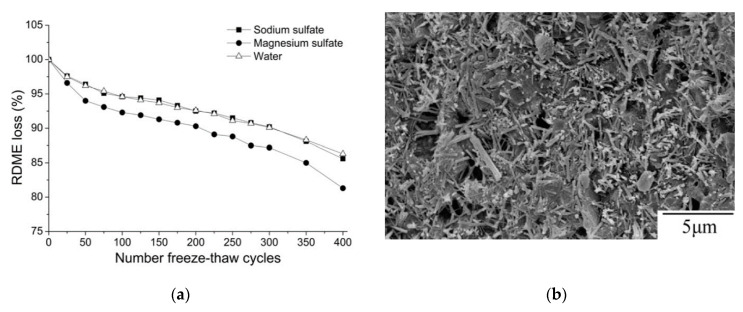
Effects of the freezing medium type on the internal damage of OPC/FA concrete exposed to a sulphate solution: (**a**) relative dynamic modulus of elastic (RDME) loss; (**b**) SEM image of needle-shaped crystals after freeze–thaw (F–T) cycles (reprinted from Jiang et al. 2015, © Elsevier Publishers).

**Figure 7 materials-13-03467-f007:**
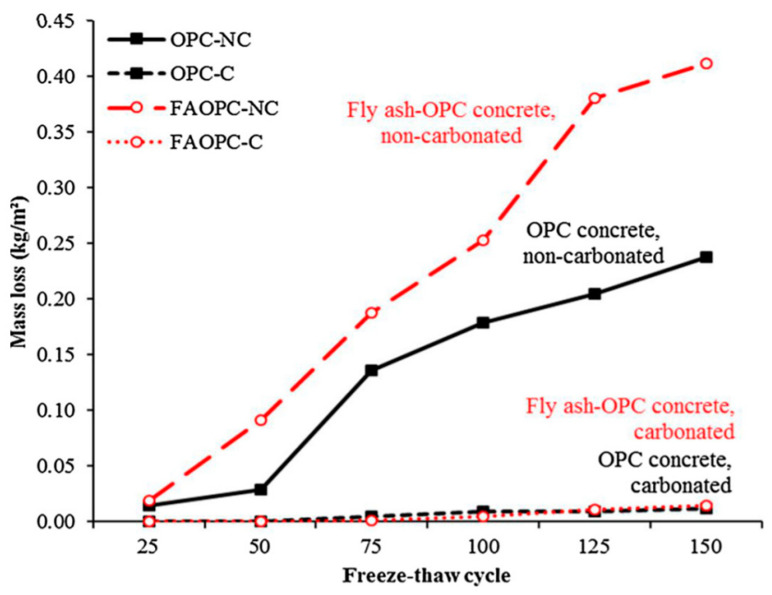
Concrete surface salt scaling vs. number of freeze–thaw cycles (carbonated—CO_2_ cured vs. non-carbonated curing (reprinted from Zhang and Shao 2018, © Elsevier Publishers)).

**Figure 8 materials-13-03467-f008:**
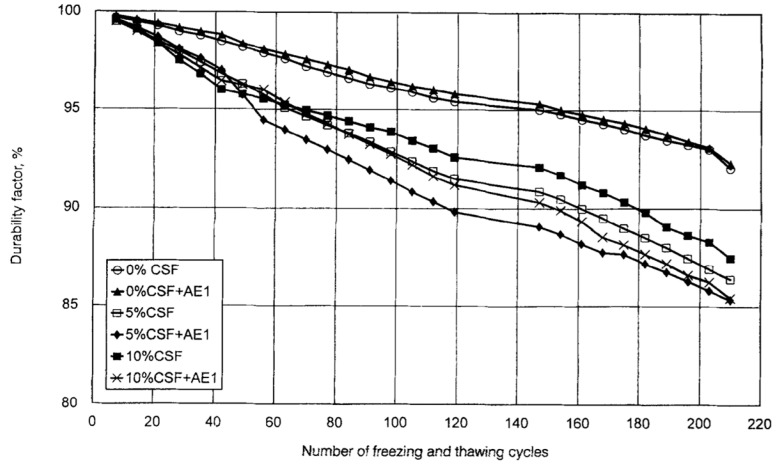
Durability factors of concretes undergoing up to 210 cycles (freezing and thawing (reprinted from Sabir 1997, © Elsevier Publishers)).

**Figure 9 materials-13-03467-f009:**
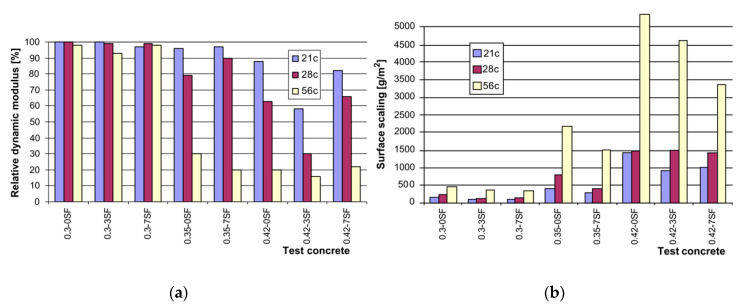
(**a**) Relative dynamic modulus and (**b**) surface scaling of concretes in the CDF test in different conditions (reprinted from Cwirzen and Penttala. © Elsevier Publishers).

**Figure 10 materials-13-03467-f010:**
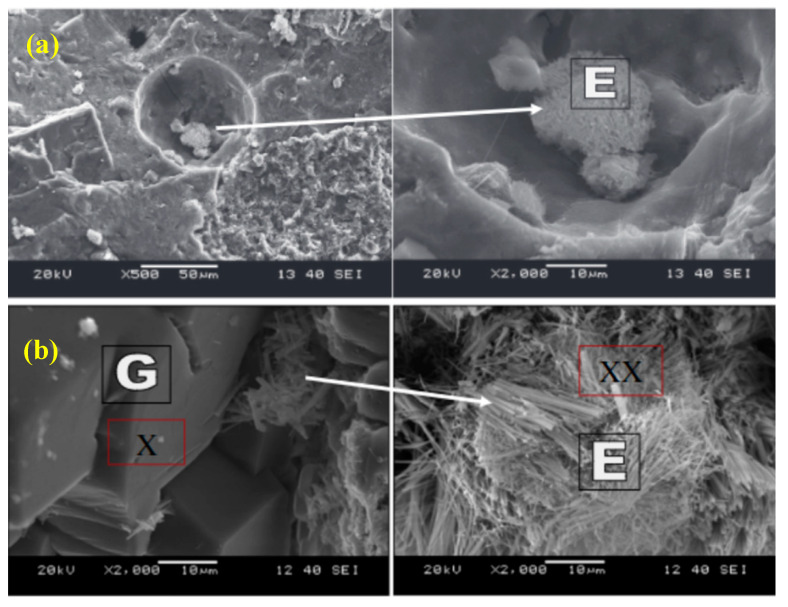
SEM images of concretes exposed to a sulfate solution, (**a**) Silica fume blended concrete (E—ettringite); (**b**) reference concrete (E—ettringite, G—gypsum) (reprinted from Mardani-Aghabaglou et al. 2014, © Elsevier Publishers)).

**Figure 11 materials-13-03467-f011:**
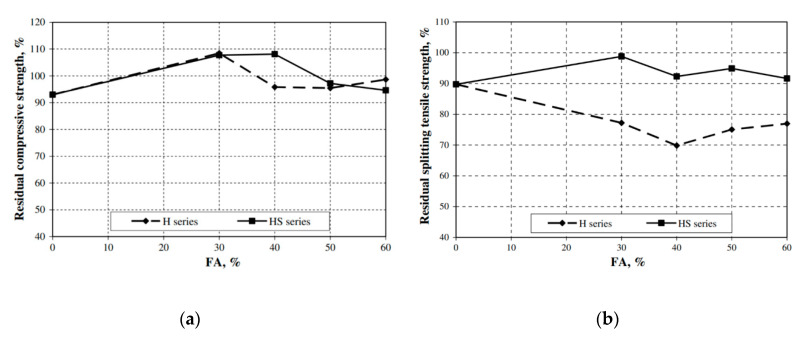
Effects of the fly ash content and silica fume addition on, (**a**) the residual compressive and (**b**) splitting tensile strength measured after 90 freeze–thaw cycles. H—reference concrete, HS—concrete containing silica fume (reprinted from Yazici 2008, © Elsevier Publishers).

**Figure 12 materials-13-03467-f012:**
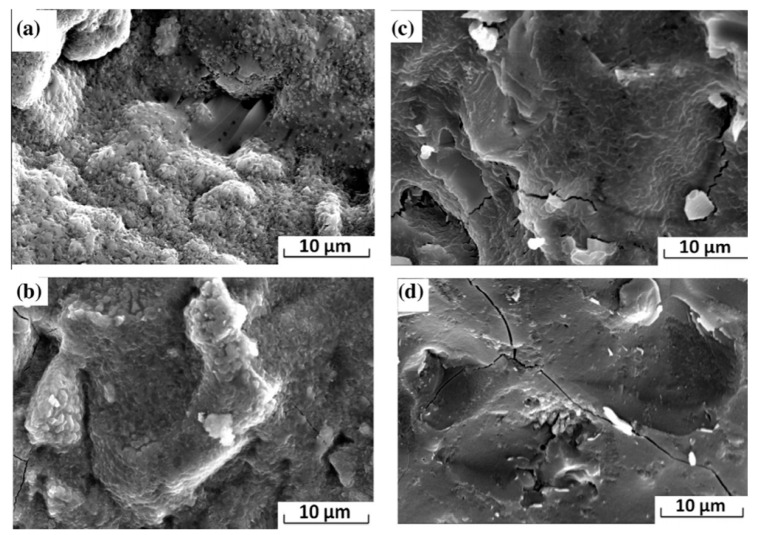
SEM of alkali-activated slag paste cured at (**a**) 7 °C, (**b**) 15 °C, (**c**) 20 °C, and (**d**) 30 °C (reprinted from Ya-min et al. 2015, © Elsevier Publishers).

**Figure 13 materials-13-03467-f013:**
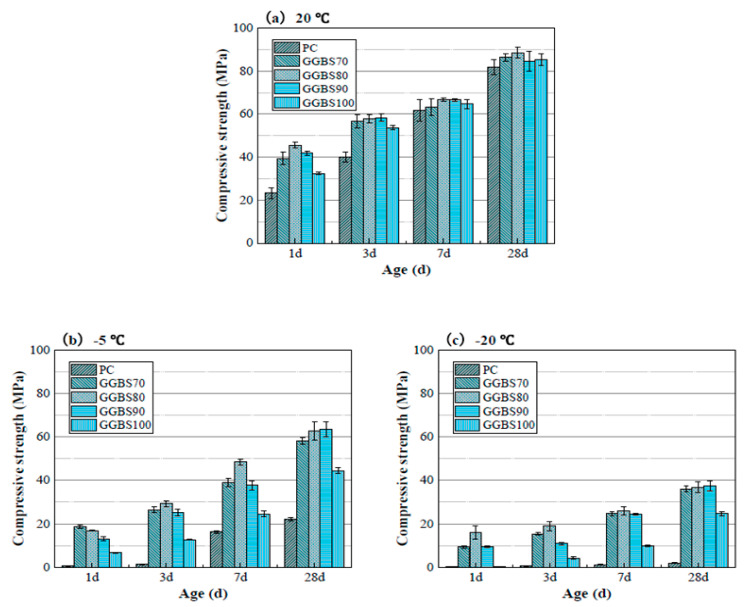
Compressive strength of binders at (**a**) 20 °C, (**b**) 5 °C and (**c**) −20 °C (reprinted from Zhang et al. 2020, © Elsevier Publishers).

**Figure 14 materials-13-03467-f014:**
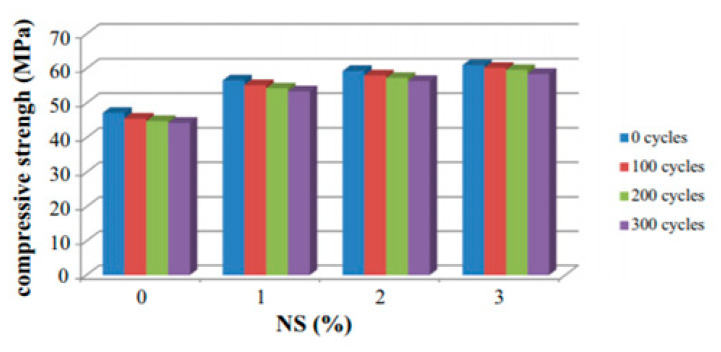
Residual compressive strength of a nano-silica (NS)-modified alkali-activated system (AAS) concrete after F–T cycles (reprinted from Shahrajabian and Behfarnia 2018, © Elsevier Publishers).

**Figure 15 materials-13-03467-f015:**
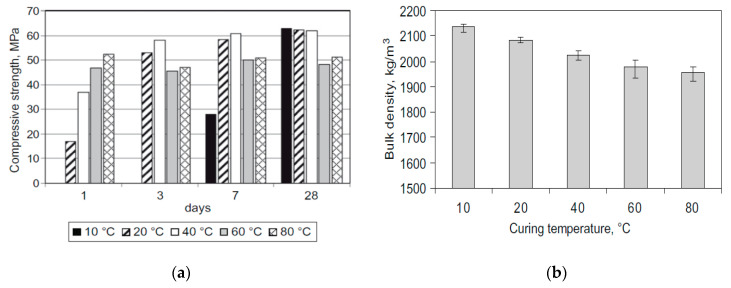
(**a**) Strength development of metakaolin-based geopolymer concrete (GPC) in different curing regimes; (**b**) bulk density after 28 days of curing (reprinted from Rovnaník 2010, © Elsevier Publishers).

**Figure 16 materials-13-03467-f016:**
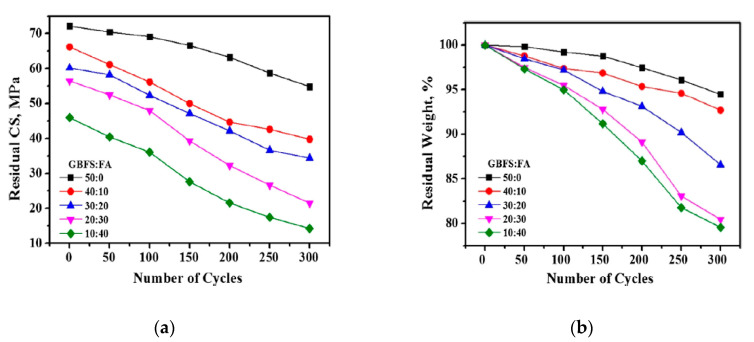
(**a**) Residual strength (MPa); (**b**) residual weight (%) after 300 freeze–thaw cycles (reprinted from Huseien et al. 2019, © Elsevier Publishers).

**Figure 17 materials-13-03467-f017:**
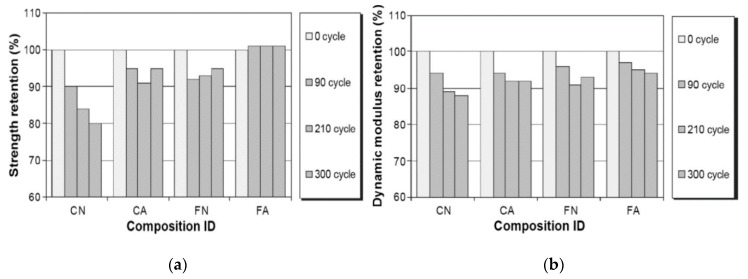
(**a**) Residual strength (%); (**b**) residual dynamic modulus (%) after repeated F–T cycles (C—cement, F—fly ash, N—no air-entraining agent, A—with air-entraining agent (reprinted from Sun and Wu 2013, © Elsevier Publishers)).

**Figure 18 materials-13-03467-f018:**
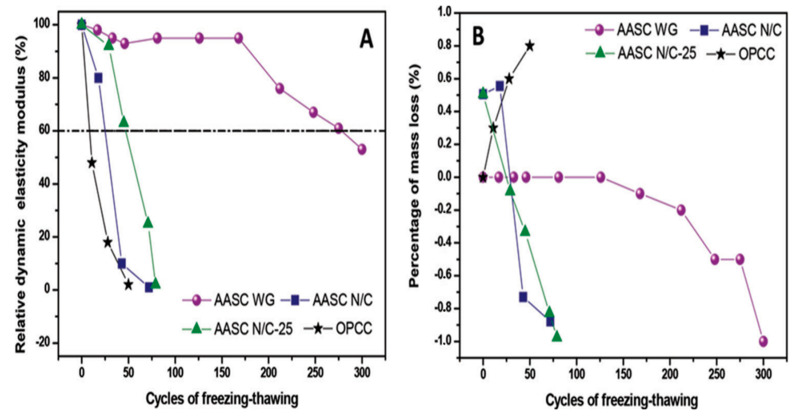
(**A**) Relative dynamic modulus of elasticity; (**B**) percentage of mass loss—against repeated freeze–thaw cycles (reprinted from Torres-Carrasco et al. 2015, © American Concrete Institute).

**Figure 19 materials-13-03467-f019:**
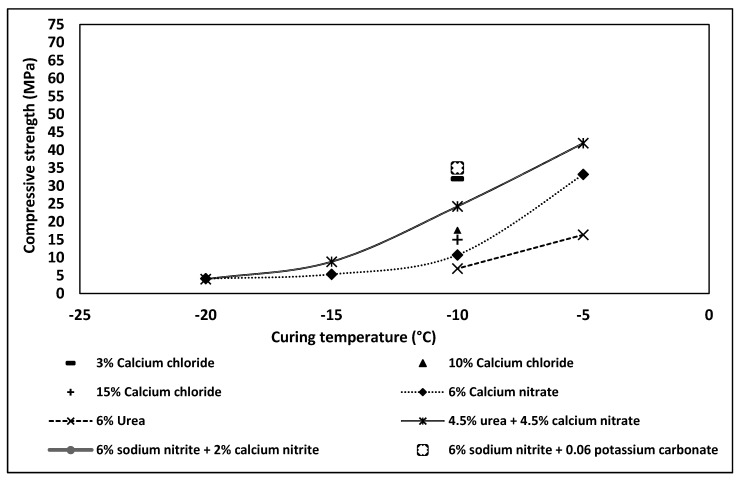
Effects of various antifreeze (AF) admixtures on 28 days compressive strength values measured for concrete based on OPC, cured at a constant subfreezing temperature.

**Figure 20 materials-13-03467-f020:**
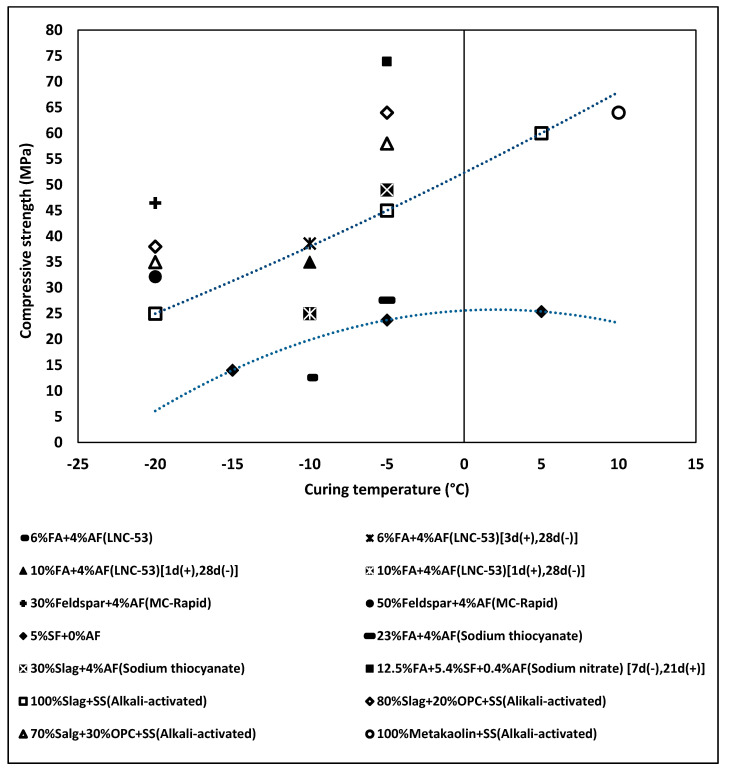
Effects of various AF admixtures on the 28-day compressive strength values measured for concretes based on composite cement and alkali-activated concrete (AAC), cured either at constant subfreezing temperatures or in variable temperatures. The curing procedure is shown in […] (indicates the curing condition enclosed in the brackets, as labelled in figure).

**Table 1 materials-13-03467-t001:** Chemical admixture for winter concreting.

Admixture	Years	Reference
Calcium Chloride	1951, 1952, 1958, 1970, 1976, 1990, 1995, 1998, 2007, 2008, 2013	[1,7,20,24,27,28,29,30,31,32,33]
Sodium Chloride	1976, 1990, 1998	[1,7,28]
Calcium Nitrate	1991, 1999, 2003, 2013, 2015, 2016, 2018	[2,20,25,34,35,36,37,38]
Calcium Nitrite	1989, 1991, 1995, 1996, 1998, 2007, 2013	[7,16,20,22,27,33,39,40]
Magnesium chloride	2008	[32]
Sodium Nitrate	1997	[41]
Sodium Nitrite	1991, 1996, 2012	[16,39]
Sodium Sulfate	1985, 1990, 1997	[1,41,42]
Potassium carbonate	1983, 1991, 2019	[16,43,44]
Urea	2014, 2015, 2016	[2,14,36]
Calcium thiocyanate	1995, 2007	[27,45]
Polyhydroxy amine	2011	[46]
Sodium thiocyanate	1988, 1998, 1999, 2003	[7,12,34,47]
Polyglycolester-based	1991, 1999	[34,35]
Hydroxyethylamine	2011, 2014	[46,48]
Polyethylhydrosiloxane	2007	[26]

**Table 2 materials-13-03467-t002:** Mix combinations of different admixtures for different sections of concrete casting (reprinted from Barna et al. 2010. © U. S. Army Engineering Research and Development Centre, Cold Regions Research and Engineering Laboratory Publishers).

Admixture	Section 1	Section 2	Section 3	Section 4	Section 5
Glenium^®^3000 (fl oz/cwt)	8	6	7	-	4.9
Pozzutec^®^20+ (fl oz/cwt)	68	34	45	34.1	22
Rheocrete^®^CNI (gal/yd^3^)	4	2.3	3	2.3	1.5
Rheomac^®^VMA (fl oz/cwt)	-	-	-	4	-

**Table 3 materials-13-03467-t003:** Type and amount of admixture for freeze–thaw analysis (reprinted from Korhonen 1990, © U. S. Army Corps of Engineers, Cold Regions Research and Engineering Laboratory Publishers).

Serial No.	Antifreeze Admixtures	% Dosage by Cement Weight
1	Conventional concrete	-
2	NaNO_2_	15
3	Ca(NO_3_)_2_ + CO(NH_2_)_2_	15
4	Ca(NO_3_)_2_/(NO_2_)_2_ + CaCl_2_ + CO(NH_2_)_2_	15
5	NaNO_2_ + CaCl_2_	15
6	K_2_CO_3_	15
7	K_2_CO_3_	25

**Table 4 materials-13-03467-t004:** Efficiency of antifreeze admixtures for OPC and composite concrete based on the available data.

Antifreeze Admixture	OPC	Composite	Comments	References
Calcium Chloride	×	-	Expansive oxychloride Susceptible to reinforcement corrosion	[1,7,24,28,29,30,31]
Sodium Chloride	×	-		
Calcium Nitrate	××××	-	Accelerates hydration reaction Efficient up to −10 °C Needs additional standard curing	[2,16,20,22,39]
Calcium Nitrite	××××	-		
Polyhydroxy amine	×××	-	Post-curing boosts strength and microstructure	[46]
Polyglycolester-based	×××	-	Depends on the dosage of admixture	[34,35]
Urea	×××××	-	Efficient with calcium nitrate Breaks hydrogen bond and enhances workability Accelerates and nucleates at the same time	[36]
Calcium thiocyanate	××	-	Accelerator	[27,45]
Sodium thiocyanate	×	× (OPC + FA) ×× (OPC + Slag)	Risk of alkali–aggregate reaction	[7,12,34,47]
Sodium Nitrite	××××	×××× (OPC + FA + SF)	Prolonged pre-curing will develop cracks	[40]
Potassium carbonate	×××	-	Detrimental effect at standard temperature	[44]
Sodium Sulfate	××	-	Need pre-curing	[43]
Hydroxyethylamine	××	-	Does not withstand corrosive environment	[46,48]
MC Rapid 25/15	-	×××× (OPC + Feldspar)	Dissolute silica compound and maintain liquid phases	[9,17]

Note: (×) not acceptable performance; (××) barely acceptable; (×××) reasonable; (××××) acceptable; (×××××) very good performance; (-) no data available.

**Table 5 materials-13-03467-t005:** Efficiency of commercial AE for OPC and composite concrete.

Air-Entraining (AE) Admixture	OPC	Composite	Comments	References
Diamidoamine salt	×××	-	Closed spaced air-void Promote strength development at later age	[62]
Saponin based	-	×××× (OPC + FA)	Stable in acidic and alkaline environment	[76,77,78,79]
Resin based	-	× (OPC + SF)	High porosity	[88,89]
Polyethylhydrosiloxane (PEHSO)	-	×××× (OPC + SF + Slag) ×××× (OPC + FA + SF)	High reactive with hydroxyl group	[26,117]

Note: (×) not acceptable performance; (××) barely acceptable; (×××) reasonable; (××××) acceptable; (×××××) very good performance; (-) no data available.

**Table 6 materials-13-03467-t006:** Effects of activators and air-entraining agents (AEs) on different alkali-activated systems (AASs).

System	Activator	AE	Rating	Comments	References
Slag	SS	√	×××	Refined air-voids	[127]
Slag	SC	√	×	Unstable microstructure	[126]
Slag	SS	-	××××	Low temperature less microcracks	[128]
Slag/OPC	SS	-	×××××	Acceleration and geopolymerization	[6]
Slag/SF	SS	-	×××××	SF complimented to fill pores	[54,130]
Metakaoline	SS	-	×××	Slow early age strength development	[129]
WSP/Slag	SS	-	×××	Increased C-A-S-H gel formation	[143]
WSP/Slag/FA	SS	-	×	FA limited the C-A-S-H gel	[143,144]
FA	SS	√	××××	AE refined pore structure	[145,146]
FA/Slag	SH	-	×××	Slag produce more alkali during thawing of F-T	[148]

Note: SS—sodium silicate; SC—sodium carbonate; SH—sodium hydroxide. (×) not acceptable performance; (××) barely acceptable; (×××) reasonable; (××××) acceptable; (×××××) very good performance; (-) no data available.

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
