# Peer review of "A Review of the Mechanical Properties and Durability of Ecological Concretes in a Cold Climate in Comparison to Standard Ordinary Portland Cement-Based Concrete"

_materials, 2020, doi:10.3390/ma13163467_

Round 1

Reviewer 1 Report

The manuscript provides a very detailed summary of the existing knowledge of the behaviour of concrete based on ordinary Portland cement (OPC) in cold climates. At the same time, it also identifies the lack of information in regard to the behaviour of new ecological concretes under such conditions.

I have the following comments on the manuscript:

  1. In my opinion, the title has not been worded accurately – the title indicates a comparison between ecological concrete and ordinary Portland cement. In fact, the comparison evidently involves ecological concrete and standard OPC-based concrete. I recommend changing (clarifying) the manuscript title. I would also recommend avoiding abbreviations in titles (OPC in this case).
  2. In my opinion, the heading of section 1.2. (Composite-based concrete) is not sufficiently accurate. I recommend making the heading more specific– e.g. “new ecological concrete” or “SCM-based concrete” or “concrete based on ecological binders”.
  3. Figure 2 is illegible.
  4. Table 8 and Table 9 are rather confusing. I recommend editing (formatting) them so that they offer better clarity. They involve a key outcome of the manuscript and the readers should be able to comprehend them without difficulty. The references to the tables are located after them, specifically at the end of the text. I recommend placing the table references before the tables themselves.
  5. The manuscript contains a number of formal errors, for example different line spacing of individual paragraphs, typographical errors (e.g. “thje” in line 572), agreement in plural forms (e.g. “a values“ in line 208, “All specimen” in line 474, “300 cycle” in line 511 etc.) and a number of inconsistencies with regard to the use of technical terminology. The manuscript arbitrarily alternates between different expressions of the same terms, for example:

freeze – thaw vs freeze-thaw

OPC based concrete vs OPC-based concrete

micro cracks vs micro-cracks vs microcracks

micro-structure vs microstructure

24hr vs 24 hours

air entraining vs air-entraining

alkali activated vs alkali-activated

Technical terms should be used consistently throughout the manuscript. (The ones in bold are those I recommend.) I also recommend a thorough proofreading of the manuscript.

Reviewer 2 Report

The manuscript is well written. I have some minor suggestions before further processing:

- I suggest to not use abbreviations in the title (e.g. OPC),

- please have a look on the following article (https://journals.sagepub.com/doi/full/10.1177/1464420713480984) when dealing with freeze-thaw durability of composite concrete mixes with fly ash,

- section 3. I suggest to discuss the finding from the article (https://www.sciencedirect.com/science/article/pii/S026382231731961X) when dealing with the phenomena of curing time of concrete with fly ash,

- The authors focused partially on highlighting the application of air entraining agents in concrete. I think that the following paper may be useful for the authors to be used in the review in case of discussing the application of air entraining agents in ecological concrete modified by waste mineral powders (https://link.springer.com/article/10.1007/s12649-018-0429-0).

Reviewer 3 Report

"Most of the alkali-activated systems have poor workability affecting the construction work". This is not correct (for example: Collins, F., & Sanjayan, J. G. (2001). Early age strength and workability of slag pastes activated by sodium silicates. Magazine of Concrete Research53(5), 321 - 326

This paper is based entirely on review of other researcher's publications. What is "new" about this research?

Reviewer 4 Report

This is quite a good review paper. However, it still needs some important improvements: Reviews should clearly identify the GENERAL TRENDS of the results . The authors almost never do this throughout the paper and they should. For a review paper, it is not enough to only show the results of each study. In fact, the authors need to show the general trend by one or couple figures for each property and discuss them. Again, this comment need to be considered for the all sub-sections in the result section. Finally, it is disappointing that the authors could not one single numerical rule/trend to portray the influence cold temperature in the concrete properties.

Round 2

Reviewer 2 Report

The required changes were made properly

Reviewer 3 Report

No Further Comments

Reviewer 4 Report

The changes that authors made into the manuscript, increased the quality of the manuscript and so it has novelty and scientific relevance for publication.